# Detection of Biological Molecules Using Nanopore Sensing Techniques

**DOI:** 10.3390/biomedicines11061625

**Published:** 2023-06-02

**Authors:** Iuliana Șoldănescu, Andrei Lobiuc, Mihai Covașă, Mihai Dimian

**Affiliations:** 1Integrated Center for Research, Development and Innovation for Advanced Materials, Nanotechnologies, Manufacturing and Control Distributed Systems (MANSiD), Stefan cel Mare University of Suceava, 720229 Suceava, Romania; iuliana.soldanescu@student.usv.ro (I.Ș.); dimian@usm.ro (M.D.); 2Department of Biomedical Sciences, Stefan cel Mare University of Suceava, 720229 Suceava, Romania; 3Department of Computer, Electronics and Automation, Stefan cel Mare University of Suceava, 720229 Suceava, Romania

**Keywords:** diagnostics, biosensors, clinical applications, ionic currents, polymers

## Abstract

Modern biomedical sensing techniques have significantly increased in precision and accuracy due to new technologies that enable speed and that can be tailored to be highly specific for markers of a particular disease. Diagnosing early-stage conditions is paramount to treating serious diseases. Usually, in the early stages of the disease, the number of specific biomarkers is very low and sometimes difficult to detect using classical diagnostic methods. Among detection methods, biosensors are currently attracting significant interest in medicine, for advantages such as easy operation, speed, and portability, with additional benefits of low costs and repeated reliable results. Single-molecule sensors such as nanopores that can detect biomolecules at low concentrations have the potential to become clinically relevant. As such, several applications have been introduced in this field for the detection of blood markers, nucleic acids, or proteins. The use of nanopores has yet to reach maturity for standardization as diagnostic techniques, however, they promise enormous potential, as progress is made into stabilizing nanopore structures, enhancing chemistries, and improving data collection and bioinformatic analysis. This review offers a new perspective on current biomolecule sensing techniques, based on various types of nanopores, challenges, and approaches toward implementation in clinical settings.

## 1. Introduction

Each year, new health conditions and their underlying mechanisms are identified as a result of continuous changes in lifestyle and due to advancements in understanding and exploring the human body. Since causes and manifestations of diseases intrinsically have molecular foundations, the detection of specific molecules involved is paramount for human health. Examples of diseases that would benefit from such detection are Alzheimer’s, Parkinson’s, or cancer, which, if not diagnosed early, may significantly reduce life expectancy and quality. An interesting and important approach in this direction is the detection of biomarkers at the single molecule level by means of nanopores [1,2].

Nanopores are structures that naturally occur as proteic polymers or can be manufactured from synthetic materials such as nanoscale silicon or graphene. When a nanopore is embedded in a dielectric membrane, it can be used to detect biomolecules, particularly DNA, RNA, and proteins [3], due to changes in the local microenvironment. Over time, various organic and synthetic nanopores, from the discovery of α-HL (α-hemolysin) organization in lipidic solutions, to solid nanopores based on silicon, graphene, etc., have been used to generate consistent information in molecular biology (Figure 1). Solid nanopores are widely used because of their flexible geometry and shapes and can be manufactured as a function of the analyte to be detected [4]. Thus, the solid-state nanopore does not undergo selective translocations and, as a result, it only performs several detections with a low degree of selectivity. Recent studies have investigated this issue and functionalized stable surfaces with recognition molecules that allow them to identify a specific molecular entity [5]. A current is established during the application of an electrical potential across a nanopore. Passing a molecule into the nanopores causes a complete or partial blockage. This blockage is characterized by the change in current and dwell time, which corresponds to the size and respective electric charge of the molecule [6,7].

Physically, the current flowing into a nanopore is a measurement of the net transport of charged loads per unit of time (Figure 2). As a result, the species entering the nanopore may influence the amplitude of the current, which is closely related to the properties of the analyte, nanopore, and solution [11]. The hydrodynamic effects (electro-osmosis), the molecular interactions, the surface of the local charge distribution (polarization, concentration of condensation ions), as well as the properties of the translocating analyte are important to determine the type of molecules that cross the nanopore [12]. Changing the properties of solid nanopores may change translocation time, which is the time it takes for the molecule to pass through the nanopore from one side to the other, independent of other experimental parameters. For example, any slight change in the superficial load may result in a significant change in the dwell time, which is the amount of time a molecule spends inside the nanopore of the analyte [13]. Thus, the passage of a molecule through a nanopore is influenced by the potential applied from the outside, the electrophysiological solution (electrolyte), and the physicochemical properties of the molecule of interest (analyte) [14]. This review aims to highlight the versatility of the nanopore analysis method and, at the same time, provide insight into the challenges the method raises, based on the molecule to be studied. The paper includes several specific experimental aspects for various molecules with important relevance for human health 

## 2. Applications in Sensing and Diagnostic

As a result of their biosensing capabilities, nanopores are the basis for the development of new technologies for the rapid detection of various specific molecules involved in several pathologies. The importance of using nanopores has been highlighted in an increasing number of studies. For example, genetic analyses have been reported to be carried out using nanopores in approximately 7 h, with patients receiving a clear diagnosis in a short time [15,16], while they are usually performed in a few days or even weeks using classical methods. 

Moreover, the sequence, structure, and components of proteins in living organisms contain information of great importance to normal or pathological physiological processes, however, numerous challenges and deficits in this field still remain [17,18]. For example, passing molecules through the nanopore too quickly presents a problem in identifying signal fluctuations that should be specific to the translocated molecule. This is why researchers explored means to slow down molecular transport. To date, there are numerous strategies that improve nanopore analysis methods by adjusting various experimental parameters to achieve high resolution [19].

However, information about the identity of an analyte is, most times, contained in the average duration of current blockages and the amplitude of blockages [20]. Each protein has a sequence of amino acids, a three-dimensional structure, and different charge profiles. This information is reflected in the detected current signal when passing through a nanopore. According to this principle, different properties of proteins or other molecules of interest in the nanopore domain have been investigated [12,21]. Nanopores are also used according to the properties of the molecule of interest. For instance, solid-state nanopores are used for large molecules. Several nanopore techniques for certain molecules are listed in Table 1.

## 3. Methodological Approaches and Challenges

The use of nanopores as a method of analysis for the diagnosis of various diseases may be an innovative method that promises to go far beyond the analyses currently performed in hospitals. The nanopore assay method promises to revolutionize both classical molecule analysis of different biomarkers and to offer the detection of molecules at low concentrations and with high resolution. The technology has become so compact that its use at the point of care represents an enormous potential [44]. One of the most important advantages of the nanopore analysis technique is real-time analysis, which offers the possibility of dynamic monitoring of biological processes [45]. However, there are a number of challenges that make this method not yet widely available. Apart from the physical and chemical optimizations required for each type of nanopore, including the structure and materials from which nanopores are made, whether synthetic or biological, there are numerous challenges in terms of molecule capture, translocation rate control, and blockages due to the analysis of molecules that are too large or that interact with the pore surface. For better resolution in nanopore experiments, it is necessary to pay attention both to the sample and to experimental parameters such as applied voltage, buffer pH, external noise, and purity of solutions. For example, the pH of the electrophysiological solution can have effects on nanopores such as in the case of OmpG nanopores which, at an acidic pH, can close [46] or influence transport processes through the nanopore [47]. The method is being fine-tuned both in terms of working time and in terms of results. With MinION™ for example, results can be obtained in about 6 h compared to 1–2 days using culture media [48]. Another factor influencing the results is the bioinformatics analysis of the signal. Bioinformatics refers to the analysis and interpretation of the resulting electrical signal to arrive at biologically or medically conclusive results, as in the case of DNA sequencing when the base-calling method is used to determine the DNA structure [49]. There are numerous algorithms that attempt to analyze the signal and use advanced methods such as machine learning, Markov models, or neural networks [50,51]. All these factors contribute to the development of nanopore analysis methods, significantly improving time-to-value ratios. At the same time, the methods need to be optimized specifically for the application. In the following sections, we discuss specific challenges and existing approaches for specific molecules of interest, to provide an overview of the field of nanopore sensing development.

### 3.1. Metabolic Markers

#### 3.1.1. Glucose Detection

Apart from diabetes, glucose is important in other physiological conditions, for example, it is the main energy source in cancer cells and, in general, glucose plays a fundamental role in providing energy for cellular metabolism. Malignant cells need more glucose than normal cells to survive, thus the amount of glucose contained in cancer cells is different from that in normal cells. It follows that accurate detection of glucose levels in individual cells is important for the diagnosis and treatment of cancers. Therefore, it would be of enormous clinical significance to distinguish the difference in glucose content in normal cells and neoplastic cells in biopsies of heterogeneous tissues, as well as to detect changes in glucose content in starved cancer cells [52]. Although methods for glucose detection in people with diabetes are widespread, as this is important for the prevention and treatment of this disease, determining glucose at the level of the individual cell is a challenge. One of the limitations in the fabrication of glucose biosensors is the charge transfer between the enzyme and the electrodes. For this problem, conductive polymers are used to facilitate the interaction between enzymes and electrodes by making charge transport more efficient [53]. Membrane glucose sensors have also been developed that have the main advantage of preventing direct electro-oxidation of chemicals in the physiological environment. The first detection of glucose without enzymes using a microchip was performed in 2006. This microchip system consisted of a network of microfluidic transport channels and a miniaturized electrochemical cell for the non-enzymatic detection of glucose. The sample and buffer were transferred to the cell by electro-osmotic flow. A Pt electrode was used to determine the glucose concentrations in the saline solution by direct oxidation of glucose without any separation process [54].

Ref. [52] Single-molecule intracellular investigations are difficult to perform due to a lack of non-destructive analytical sensors and controlled cell-targeting technology. Due to the small pore diameter, a technological solution such as a nanopipette can be introduced into individual cells repeatedly, as the tip is less than 100 nm [55]. Bio-recognition materials such as antibodies that selectively bind to specific cells can be inserted at the tip of the nanopipette [56,57]. Single-molecule glucose detection is a diagnostic tool in research for distinguishing between malignant and benign cells in biopsies and a tool for monitoring cancer progression in situ. The research seeks to automate the system to increase single-molecule analysis efficiency to 120 cells/hour [58].

#### 3.1.2. Vitamin B1

B-complex vitamins are essential for both the physical and mental state, having important functions in the cognitive, digestive, muscular, and cardiac systems, and playing an important role in energy metabolism [59]. Vitamin B deficiency can lead to mild symptoms such as fatigue but can also result in permanent neuronal degeneration and significant disorders in the human body [60]. Vitamin B1 (Thiamine) molecules are distinct from the other molecules in the B complex in that they contain a thiazole and a pyrimidine ring [59,61]. The determination of vitamin B1 status is usually performed in medical laboratories by various techniques, such as liquid chromatography [62] and UV-VIS spectrometry [63], within a few hours. The nanopore technique also tends to make the methods more efficient in terms of time. To this end, a biological nanopore of ClyA (Cytolysin A) has been used to determine the concentration of thiamine in urine using automatic data analysis that takes less than 1 min [26]. The particularity of this technique consists in attaching to the nanopore a protein that has the property of binding to thiamine when translocated through the nanopore. By replacing a residue (Y27A), the affinity between TbpA (periplasmic thiamine-binding protein) and thiamine is increased [64,65], which improves the signal resulting from the translocation of the molecule through the nanopore, allowing the identification of thiamine.

#### 3.1.3. Uric Acid Detection

Uric acid (UA) is the main product of the breakdown of the metabolism of purine nucleotides (e.g., adenine, guanine). High serum uric acid levels are associated with many diseases, such as gout, arthritis, kidney disease, and others. Current clinical tests such as the colorimetric method or enzymatic method used to measure uric acid in urine and serum involve the use of uricase enzymes and other specific reagents [66,67]. Enzymatic methods suffer from interferences caused by chemical reactions taking place, and the analysis requires careful pH control. Therefore, it is worth developing reliable and inexpensive non-enzymatic methods for determining uric acid levels. The glass capillary nano-pipette method is a non-enzymatic strategy for the determination of uric acid. This involves a glass capillary nanopore over which an ultrathin gold substrate is inserted by chemical reduction of HAuCl_4_ on a poly (L-histidine) nanoshell attached to the substrate in the presence of NH_2_OH-HCl [28]. This nanopore is functionalized with 2-thiouracil by self-assembly and tested for non-enzymatic detection of uric acid. This method of uric acid detection is selective and reproducible. In addition, such a glass nanopore could be used for the practical analysis of serum samples. A detection limit of 1.9 μM was observed in the proposed experimental setting and the concentration in serum samples detected in hospital laboratories was 360 μM [27].

### 3.2. Detection of Nucleic Acids

#### 3.2.1. Bacterial Pathogen Identification

The detection of pathogenic bacteria is important to ensure adequate responses in clinical settings, such as ICUs (Intensive care units), for timely detection of nosocomial infections and catheter infections but also in quality assurance in the food industry. To avoid pathologies caused by bacteria, the process of detection and identification is essential [68]. Hospital infections are the cause of many deaths, for example, bacteria that cause sepsis in plasma can now be detected by microbiological culture which takes 1–2 days. For every hour that the administration of the specific antibiotic is delayed, the risk of the patient dying increases. More specifically, if appropriate therapy is not started within 1 h of shock, the risk of death increases from 1.67 in 2 h to 92.54 for delays > 36 h [69,70]. During this time, patients receive broad-spectrum empiric treatment to cover as many action spectra as possible, which can lead to multiple side effects and can also lead to antibiotic-resistant bacteria. Rapid detection of pathogens is required for early administration of antibiotics, thus avoiding the aforementioned problems resulting from diagnostic delays. Recent research that focused on Oxford Nanopore Technology sequencing (ONT) showed that detection of bacteria is possible within 5–6 h of sample collection, thus enabling much faster diagnoses [48], possibly even in clinical settings. The Nanopore technique was compared with classical techniques (Illumina) to identify the influenza A virus directly from respiratory samples. The data showed that the read ratio generated by Nanopore and Illumina metagenomic sequencing is very similar, therefore the detection limit is probably very close for these two platforms [71]. For example, when comparing 15 samples of IAV (Influenza A viruses) with 16 HMPV (Human Metapneumovirus) samples at a sequencing depth of 10×, the agreement between the two methods (Nanopore and Illumina) was 99.95%. In addition, the high error rate of nanopore reads is compensated by the increased read length, which could provide an advantage in the taxonomic assignment of individual reads [72].

With the spread of antibiotic-resistant pathogens and the associated increase in human morbidity and mortality, antibiotic-resistant genes have been identified as emerging environmental pollutants by the World Health Organization [73]. Oxford Nanopore Technology provides rapid real-time sequencing and has been used for the rapid detection of pathogens, including the diagnosis of bacterial meningitis, bacterial respiratory tract diseases, infective endocarditis, pneumonia, and other bacterial infections. The method is also very useful for detecting antibiotic-resistant bacteria or other mutations resulting from antibacterial treatments. As bacteria develop new antibiotic resistance mechanisms, this causes significant problems in the treatment of infections. Because the use of broad-spectrum antibiotics may generate antibiotic resistance, accurate and rapid detection of both bacteria and antibiotic resistance is essential in treating infections [74]. For example, the MinION™ has been quite successful in identifying pathogens. In the case of bacterial meningitis, 16S amplicon sequencing took only 10 min, and in the case of some respiratory tract-influencing bacteria, MinION™ analysis was performed in 6 h, compared to 2 days using the culture method [75,76]. Nanopore Targeted Sequencing (NTS) is an accurate and rapid analytical method for the detection of infection-causing pathogens found mainly in body fluids. Another comparison between classical methods and nanopores: the culture, NTS, and Sanger sequencing methods demonstrate the accuracy and precision of nanopores. When samples from patients with and without infections were tested by classical and NTS methods, 290 out of 307 (94.5%) patients with infectious diseases were diagnosed with NTS, while routine culture determined 116 out of 307 (37.8%) infectious diagnoses [77]. Another study carried out in 2020 confirms the results presented above, as 58.7% of specimens were detected by the NTS method, compared to 22.1% detected by the culture method [78].

Nanopore-based sequencing technology offers the ability to produce long reads from individual molecules in real time, with the potential to introduce new sequencing methods with clinical applications. Since its commercial launch, ONT has suffered from a high error rate that had to be mitigated by experimental methods, computational corrections, or signal filtering. Owing to many updates in the chemistry tools and software, the raw read accuracy has steadily increased from 60% to 85% and beyond. At the moment, the data produced by MinION™ is accurate enough to sequence with an accuracy of > 99% [79], as stated in the manufacturer’s manual: “In addition to high single-pass accuracy, the nanopore platform is now able to deliver single molecule consensus accuracy of 99.995% with UMIs, and Q50 (99.999%) consensus sequencing data” [80]. The quality (Q) of the assembled genome depends not only on the chemical corrections and filters applied but also on the properties of the sequenced molecule, for example in the case of rare species (e.g., *Proteus vulgaris* Q27 to Q32), the degree of polishing is lower than in common species (e.g., *K. pneumonia* Q26 to Q38) [81].

Pathogen detection might be more prone to interference from antibiotics in conventional culture testing, while NTS is much less affected. Some bacteria with specific nutritional requirements are difficult to cultivate even on supplemented medium or are slow-growing. As such, several bacterial species in normal and pathogenic microflora are unculturable, which poses a challenge to their detection. For these species, sequencing has proved a viable method, which may allow precise diagnostics, whereas Nanopore sequencing, due to fast protocols and sequencing times, may further provide rapid tools, even in clinical settings. Thus, NTS plays an important role in the diagnosis of infectious diseases, especially after antibiotic treatment or infections caused by slow-growing or precocious microorganisms. A rapid and accurate method such as NTS would contribute to the early administration of targeted treatment and reduce the use of broad-spectrum antibiotics. Classical culture media have a response time of about 24–48 h while in the case of NTS, the time to obtain a result varies from 6–18 h, thus the method is much more reliable with an acceptable cost since a rapid determination of the pathogen would reduce treatment costs and duration of hospitalization [82].

#### 3.2.2. Genetic Markers

A biomarker is a characteristic indicator of a specific biological condition, which can be associated with a disease. For example, the identification of biomarkers provides information about the process of change in cancer cells [83].

MicroRNAs have been investigated as potential biomarkers, and although they are relatively short sequences of about 18–22 nucleotides which should make sequencing more accessible, detection of microRNAs in the bloodstream remains difficult. The results also show that the nanopore HL sensor can detect microRNA structures from plasma samples taken from lung cancer patients without the need for label addition or sequence amplification. The method is based on the difference in concentration of the electrolyte solution between the cis and trans parts which favors the flow of ions under the influence of the concentration gradient. This is also observable at low concentrations. The technique facilitates translocation at concentrations down to 1 pM [33]. In the case of solid (glass) nanopores, faster translocation is based on the dielectrophoretic force that is applied when opening a nanopore, a technique that appears to be more efficient, with translocation occurring up to concentrations of 5 fM [84]. It has subsequently been shown that, in the early stages of cancer, the concentration of microRNA is in the femtomolar range. In fact, in early-stage lung cancer, the microRNA concentration is about 1 fM, which makes the newly developed methods more sensitive and faster. Molecule analysis at concentrations of 1 fM was achieved by combining oligonucleotide amplification and the asymmetric method [85]. In recent studies, microRNAs have also been detected at concentrations of 0.5 fM in plasma samples, demonstrating the versatility and accuracy of the nanopore. Improved mechanisms increase the detection sensitivity of nanopores which will contribute in this case to the characterization of liquid biopsies [86].

Nanopore sequencing allows for variant detection, as shown for six SNPs at different loci, with the potential for assessing perioperative outcomes [87] or even insertions and deletions several base pairs long [88]. A hallmark of genomic instability that has a major influence on cancers is so-called structural variants (SV), which include structural changes such as translocations, amplifications, or inversions. The ability of methods using nanopores to read through repetitive regions could make them an ideal tool for detecting structural variants associated with tumors [89]. In pancreatic cancer, the tumor genes *CDKN2A/p16* and *SMAD4/DPC4* are difficult to detect, given the transformational processes they go through such as inversions or translocations, which virtually inactivate the tumor genes. Using PCR amplicon mixtures, it has been shown that the nanopore sequencing technique can detect translocations, inversions, and dilutions up to 1:100, with only 500 reads per sample [89,90].

In the case of breast cancer, for example, genome-wide sequencing analysis of DNA methylation (which involves long reads) would provide important clinical information. An improved variant of nanopore sequencing for this context is based on enzymatic base conversion, where an unmethylated cytosine should be converted to thymine. This type of analysis is reliable for a minimum of 1 ng genomic DNA, obtaining a read length of 3.4–7.6 kb [91].

mDNA (DNA methylation) detection is an essential process in the analysis and control of gene evolution. Methylation detection techniques such as bisulfite sequencing [92] and MeDIP-seq (Methylated DNA immunoprecipitation sequencing) [93] are widely used. However, they can be time-consuming and expensive. Nanopore sequencing technology offers a real-time alternative. The advantage of the technique is that it can facilitate the detection of methylation in long repetitive regions and can also detect methylation in low-volume samples [94]. Unlike classical methods, methylation detection with nanopores can be bioinformatically challenging, requiring specialized expertise. Although the nanopore methylation analysis method is not widely used, it is a promising route to real-time, high-throughput detection. Since the nanopore technique can correlate the values of the electric current intensity with the properties of the molecule, the identification of methylated genes is performed on the same principle, observing differences between unmodified and modified bases [95]. It is not enough just to identify differences in current amplitude, for a satisfactory yield, it is necessary to directly call the modified bases, and for these different algorithms are used [50], such as the Markov model [96] and neural networks [97,98]. Currently, there are multiple software tools developed by researchers for this purpose, but most of them do not fully satisfy the requirements as the methylation levels are not linearly distributed [99].

An example of a practical application is the change produced by the active substances in cigarette smoke on DNA profiles, effects which have also been observed in passive smokers or even in newborns who were exposed during pregnancy [100]. The study was carried out on 9389 subjects, of which 2433 were current smokers. The results showed that 2623 of those monitored had changes in DNA methylation. A CRISPR-Cas9 targeted sequencing with Nanopores was used as a method to determine the mDNA sequences. The method showed satisfactory results, identifying 1779 CpG regions. These were associated with three genomic regions that are known to be related to smoking [101]. CRISPR-Cas is used for rapid and sensitive detection of DNA or RNA targets. A combination of CRISPR-Cas-based bioanalytical systems and nanopore sequencing can be performed for better accuracy, with CRISPR-Cas specifically enriching DNA sequences for better and rapid nanopore sequencing. In the meantime, CRISPR-based sensing tools can accurately recognize specific sequences or molecules and perform electrochemical tag modifications, thus changing currents through the sensing device [102,103]. Another direct approach to identifying 5-methylcytosine (5mC) highlights the selectivity of nanopores. The method is based on the interaction of the modified cytosine with anionic acids in biological nanopores without any further chemical modification of the DNA [104].

### 3.3. Protein Detection

Proteins are present in all living organisms and play an important role in the structure and function of cells, with proteins made of up to thousands of amino acids. Identifying the properties of proteins involved in different pathologies would lead to faster and more efficient diagnosis [105,106]. However, proteins are more difficult to characterize than DNA, firstly because of the three-dimensional structure of proteins, the dimensions of which are comparable to, or larger than, the diameter of nanopores. Secondly, proteins are non-uniform charge-carrying structures due to having a large number of amino acids which makes translocation through a nanopore even more difficult compared to DNA which has a uniform charge distribution along the length of the molecule [107]. To overcome these difficulties, other approaches such as dielectrophoretic capture and electro-osmotic flow have been studied [108]. It has been demonstrated that, in certain conditions, such as neurological disorders (Alzheimer’s, Parkinson’s, and Prion diseases), the proteins involved undergo structural changes. Early diagnosis of these diseases can improve quality of life. For example, an increasing number of studies validate the accuracy of nanopore sequencing for rapid diagnosis of prion disease and highlight the need for single-molecule sequencing methods for somatic mutation detection. The study established a new long-read sequencing strategy, succeeding in reading the full length of the PRNP (Prion gene) of 16.7 kb [109]. However, the technique presented can be improved, since it currently requires amplification of PRNP in two fragments. If the experimental procedure was simplified, PRNP detection could be done in the field. Researchers have considered using a targeted sequence, for example, CRISPR-Cas9 [110] or the “read to...” mode of Oxford Nanopore devices [111]. Among the diseases that irreversibly affect neurological functions, Alzheimer’s disease affects 6.2 million Americans, a number estimated to double in the next 30 years [112]. Symptoms of Alzheimer’s disease progress over time, which is why early detection of disease triggers is essential. It is known that the aggregation of Aᏸ peptides is the cause of this disease, therefore the determination of the factors that favor aggregation is desirable [113]. The use of nanopores to detect changes in Aᏸ 40 sequences is an important step since nanopores have the sensitivity to identify conformational changes [113]. The level of quantification attainable using molecular sensing is not achieved by classical spectroscopic, gel electrophoretic, or ELISA-based detection techniques [114]. Techniques using nanopores propose a simple way to simultaneously obtain information about the size, shape, and distribution of heterogeneous protein aggregates using a small volume of solution [115]. However, this is not without challenges due to the complex structure and specific electrical charge of the protein. One way to improve protein detection is to slow down their movement to increase the dwell time in the nanopore [116]. Since the charge of the protein influences the translocation process through the nanopore, one solution is to work at a pH close to the isoelectric point of the protein [12]. This makes the net charge of the protein molecules almost zero, consequently, electrophoretic mobility is decreased, which slows down the protein translocation [117]. Another way to slow down the passage is to modify the surface properties of the nanopore. For example, the SiNx nanopore can acquire a slightly negative charge at a given pH or under the influence of light intensity. This factor can alter the translocation process [116,118]. Another method is to bind a ligand to the protein. For example, DNA chains containing aptamer sequences have been linked to specific target proteins. Protein detection was determined by analyzing the amplitude of the ionic current when DNA chains translocated through the pore [119]. The use of the Monte Carlo method to introduce a probabilistic model using aptamer-functionalized nanopores is intended to identify proteins in serum at pM concentrations. In this regard, studies examined the identification of thrombin at a concentration of 50 pM in serum. The technique has been verified over 25 repetitions, and the results have shown lower detection limits, allowing the platform to advance toward clinical testing [120].

Another marker is αS (α-synuclein), which is a biomarker of synucleinopathies (Parkinson’s disease, Lewy body dementia, multiple system atrophy), and rapid detection from a small amount of the substance is ideal for early diagnosis. A nanopipette combining in a single system a reaction vessel for accelerated amplification and a nanopore sensor allows the detection of αS assemblies in a given sample in only 90 min by adding a small amount (35 μL at 100 nM) of recombinant αS for amplification [121]. A rapid and correct diagnosis is critical to increase the chance of life and maintain quality of life. In this respect, nanopores are a promising technology that can help. For example, cancer is one of the diseases with a more or less chaotic evolution that requires early and precise identification of biomarkers that are involved in the decline of the disease. Moreover, new tumor entities have been identified that cannot be characterized histologically, leading to the need to characterize cancers molecularly. For example, the World Health Organization explicitly requires molecular testing for central nervous system tumors. However, the implementation of integrated histomolecular testing requires a multitude of technologies, and the time to implement it can be several weeks, making standardized implementation difficult [122]. Things can also be improved in cancer cases that already have an established diagnosis and treatment protocol. For example, in the case of prostate cancer, patients with high levels of PSA (Prostate-Specific Serum Antigen) undergo painful and expensive procedures such as biopsy. The test can be improved by finding a protein that is selectively secreted by malignant cells. To this end, researchers have discovered a protein called NEM (neuroendocrine marker) that is secreted only by malignant prostate cells and released into the bloodstream [123]. Improved systems that make the nanopore technique reliable in clinical testing are increasingly being tested. For example, one system that can measure NEM and PSA consists of an optofluidic chip made up of nanopore-based sensors made of anodic aluminum oxide. The results show that the chip had improved sensitivity, requiring only 1 μL of serum to measure PSA and NEM. In addition, compared to ELISA tests for PSA, the nanopore-based technique is 50–100 times more sensitive [38].

Another important protein, hemoglobin, is an essential, iron-containing protein found in red blood cells. Its function is to carry oxygen from the lungs to every cell in the body and to take the carbon dioxide resulting from metabolic burning to the lungs for disposal [124]. Folded hemoglobin can also be detected in blood samples by techniques using nanopores, for example, PlyAB (pleurotolysin) biological nanopores show promise for the real-time analysis of folded proteins [40]. It has been possible to identify hemoglobin even in variants in which a single amino acid differs, representing an accuracy of 97%. The concentrations identified as the normal range is between 15–17 g/dL (2.3–2.6 mM), and the nanopore is able to identify proteins even in concentrations of 1 mg/dL (150 nM). Therefore, the use of nanopores for automated analysis of blood samples would be useful in the identification of hemoglobin variants and thus in the diagnosis of specific diseases caused by changes in hemoglobin levels [125]. Likewise, the presence of insulin antibodies is a predictor of type 1 diabetes for people who do not have diabetes but are genetically predisposed. Early identification of this factor helps to prevent and manage diabetic disease accurately [126]. To facilitate the passage of insulin through a solid nanopore, the system must be adjusted to slow translocation. Thus, the nanopore with a metallic film was fabricated by electron beam sputter deposition of Au on a SiN_x_ layer. Finally, the Au surface was modified with Hcy (homocysteine), which led to a significant change in the current-voltage characteristics, practically after the modification the translocation time of insulin through the pore increases, which is explained by hydrophobic interactions between the protein (insulin) and the Hcy layer [127]. The use of homocysteine-modified metal nanopores has been shown to be effective, as protein uptake on chemically modified nanopores slows the translocation process to tens of milliseconds [128]. Another approach to the analysis and subsequent sequencing of proteins is tried with the ONT method, which is now used for DNA sequencing. There are a few articles using the ONT method for protein analysis, one of the most relevant uses an R9.4.1 cell and a specific modified script and protein tags (NTERS). A negatively charged protein (NTERY00) is used to develop this method. The negative loading of the C-terminal end facilitates the capture of the protein by the nanopore. Protein analysis with MinION™ using NTER barcode engineering is promising as the method reveals both protein expression and specific post-translational modifications [129].

## 4. Protein Sequencing

Single-molecule protein sequencing would greatly impact detecting and treating rare diseases or diseases involving a specific protein such as Alzheimer’s. For example, the difficulty of sequencing using biological nanopores is the small inner diameter, making it almost impossible to pass the secondary or tertiary structures of proteins, requiring protein blunting processes. In fact, the protein must spend a certain detectable time inside the nanopore so that the events taking place can be analyzed. The latest research in the field focuses on single-molecule protein sequencing, and studies can now be carried out that target interaction recognition, structure characterization, etc. [130] Although for DNA sequencing, the recognition of the four deoxyribonucleotides has been achieved, the sensitivity of biological nanopores is not sufficient to identify 20 amino acids. The use of the aerolysin (AeL) pore in protein sequencing seems to be the most optimal option. The aerolysin detection zone allows the protein to stay inside the pore for up to a few milliseconds, which facilitates the analysis of the blocking time and thus the electrical current amplitude [131]. However, progress is being made in single-molecule protein sequencing. For example, using an AeL nanopore, six peptides have been identified that differ in a single amino acid. Current fluctuations were regulated by changing the pH [132,133]. Moreover, different variants of the aerolysin nanopore can be associated with synthetic nanopores, resulting in so-called hybrid nanopores. This type of nanopore is expected to combine robustness and sensitivity to facilitate protein translocation [134]. Solid-state nanopores have been used for short protein sequencing. In order to read whole proteins, unfolding of the proteins is required, which involves denaturation. Solid-state nanopores have the great advantage of being stable and not undergoing changes due to chemical processes of protein denaturation. Moreover, temperature also enhances the denaturation process but increases the translocation rate, which prevents the detection of specific events. Identifying the optimal single-molecule protein sequencing technique would revolutionize proteomics research. In addition, identifying proteins at low levels in biological fluids would allow the identification of pathology-specific biomarkers [135].

## 5. Emerging Nanopore Technologies

Single-molecule DNA sequencing has come to be integrated into an efficient device such as one from Oxford Nanopore Technologies. However, single-molecule protein sequencing technology processes are more difficult, mainly due to a large number of amino acids, the unique nature of each polypeptide, which cannot be replicated as with DNA, and also the movement of non-uniformly charged proteins [136,137]. Plasmonic nanopores have recently been developed and are distinguished by the electromagnetic field projected around a nanoporous sensor which allows for improved optical spectroscopy, temperature control, etc. [138]. Future research directions are multiple, for example in the case of plasmonic nanopores it is envisaged to use other device media such as 2D materials to facilitate high-resolution detection, or to use plasmonic nanopores as filtering devices considering correct surface functionalization. In terms of denaturation and linearization of a protein molecule to facilitate translocation, plasmonic heating can be used [136]. Biological aerolysin nanopores are extremely versatile, being suitable for peptide and genomic detection due to their high sensitivity and specificity [139]. Moreover, aerolysin nanopores have the ability to accurately read digitally encoded information, making the detection limit no longer a barrier [140]. In order to improve the reading accuracy, the techniques have to be adapted according to the pore and environmental chemistry, the type of molecules to be studied, the base calling algorithm, or different external factors [141]. The most optimal choices will lead to an accurate and reliable nanopore sequencing method.

## 6. Conclusions and Perspectives

Nanopore sequencing is of major interest to the future of medicine. In order to make the transition from experimental variants to clinical routine, simplified and reliable working methods are needed that can be performed quickly and at a reasonable cost. Nanopore-based technology appears to be one of the newest and most promising methods for single-molecule analysis. Protein pores of biological origin are used mainly because of their selective properties for sequencing DNA, RNA, proteins, and metal ion molecules. 

On the other hand, solid nanopores are easier to handle due to their flexible geometry, being manufactured according to their use. Such pores detect only the ionic current that occurs at a given time in the system due to the translocation of the molecule. The optimization of the type of nanopore used, correlated with the molecule of interest and with experimental parameters, can lead to a new, reliable technique that can later be implemented for accurate and rapid clinical diagnostics. The advantages of these techniques include long read lengths, low substrate volume, direct identification of base changes, direct RNA sequencing, and portability. Nanopore methods are increasingly complex and can be tailored to the specific molecule, which is why expanding the knowledge and research will provide the basis for developing new techniques. Here an attempt has been made to bring to the forefront some of the methods using nanopores, aimed at rapid and efficient diagnosis of the most common and current diseases, amplifying the importance of developing and improving current methods. The versatility of using nanopores in the context of various molecules of significant importance in human health is also noted, highlighting the challenges that this method raises, as well as the ability of nanopores to analyze in real-time at the single-molecule level. Techniques using nanopores have comparable accuracy to the classical methods that are currently used, however, the time in which the sequencing result is provided is significantly shorter, thus finding the best methods for a fast and accurate diagnosis is of significant interest. The future of nanopore technology holds great promise in a variety of fields, starting with genomic sequencing, where significant progress has already been made, disease diagnosis, single molecule analysis, and even in synthetic biology and nanomedicine. The improvement of the technique should focus first on reliability and accuracy, but also on aspects such as miniaturization and portability, improved speed, integration with other technologies, and detection and analysis of new molecules.

## Figures and Tables

**Figure 1 biomedicines-11-01625-f001:**
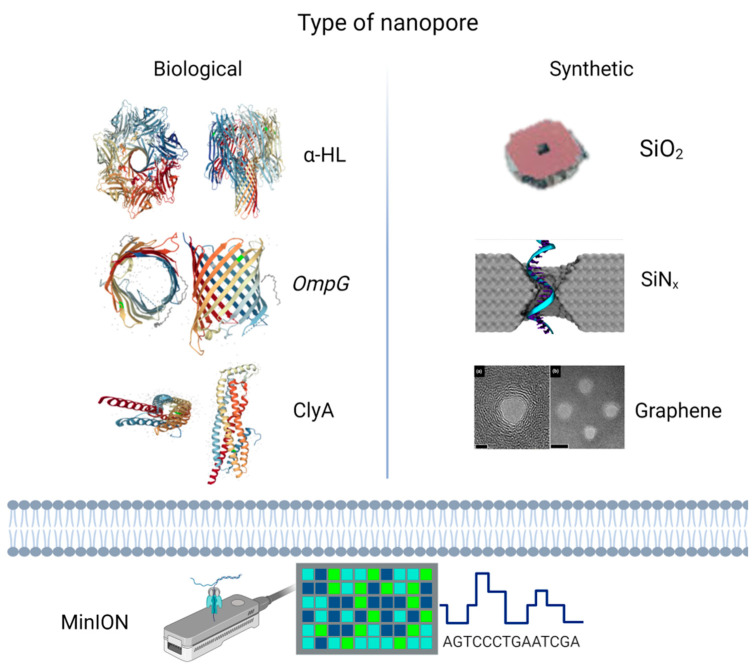
Types of nanopores with examples of biological nanopore structures (**left, based on SWISS-MODEL repository**) and microscopic representations of synthetic nanopores (**right**), adapted from refs. [8,9,10]. ONT’s commercial technique—the MinION™ (**bottom**). (Created with BioRender.com accessed on 15 May 2023).

**Figure 2 biomedicines-11-01625-f002:**
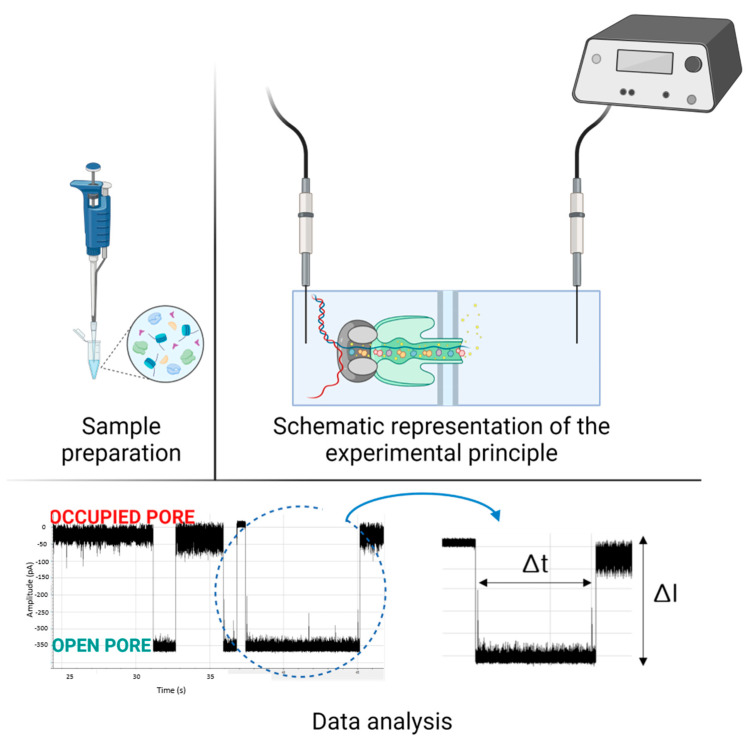
Schematic representation of the nanopore experimental protocol. First, the sample is prepared, then introduced into the nanopore system which is connected to an amplifier and the acquisition system. After this, the data analysis follows, the two states of the nanopore (occupied pore—I = 0 pA)/open pore—I) and the duration of such an event are highlighted. (Created with BioRender.com accessed on 15 May 2023).

**Table 1 biomedicines-11-01625-t001:** Examples of biomarkers with implications in different pathologies or physiological conditions that can be detected with nanopores.

	Molecule of Interest	Disease/Physiological Status	Experimental Setup	Reference
Metabolic markers	Glucose	Diabetes, cancer	Glass nanopore based on nano pipettes	[22,23,24,25]
	Vitamin B1	Normal functioning of the human body	ClyA	[26]
	Uric acid	Gout, arthritis, kidney disease	Glass capillary nano-pipette	[27,28]
Detection of nucleic acids	Bacteria	Infectious diseases	α-HL, silicon nanopore, MinION™	[29,30,31,32]
	Genetic markers: microRNA, pancreatic cancer marker, breast cancer marker	Cancer	Biological and solid-state nanopores	[33,34,35,36,37,38]
Protein detection	Prion gene, ᏸ-amyloid, α-synuclein, PSA, insulin, hemoglobin	Neurologic diseases, cancer, autoimmune diseases, etc.	α-HL, ClyA, aerolysin, SiN, graphene Al2O3, SiO2, PlyAB	[17,39,40,41,42,43]

## Data Availability

Not applicable.

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
