# Peer review of "Detection of Biological Molecules Using Nanopore Sensing Techniques"

_biomedicines, 2023, doi:10.3390/biomedicines11061625_

Round 1

Reviewer 1 Report

The paper "Detection of Biological Molecules Using Nanopore Sensing Techniques" provides an overview of the applications, methodological approaches, and challenges associated with nanopore sensing. Overall, the paper provides a good foundation on the topic of nanopore sensing. By addressing below suggestions, the paper can be further improved to enhance clarity and provide a more comprehensive understanding of the subject matter.

1.      Introduction:  Consider providing a clearer motivation for the importance of nanopore sensing in the introduction. Include more recent references (beyond [3]) to showcase the latest advancements in the field.

2.      Applications in sensing and diagnostic: Provide more specific examples and references to support the claim that nanopores are the basis for the development of new technologies for rapid detection of various specific molecules involved in pathologies. Expand on the advantages and limitations of nanopore analysis methods compared to other techniques currently used for molecule detection.

3.      Methodological approaches and challenges:  Clearly define and explain key terms such as "translocation time" and "dwell time" to ensure better understanding for readers. Include specific details about the experimental parameters that can be adjusted to achieve high resolution in nanopore analysis methods. Provide additional references to support the discussion on bioinformatics analysis of the signal and the algorithms used for data analysis.

4.      Specific molecule detection: Organize the subsections under "Metabolic markers" (Glucose detection, Vitamin B1, Uric acid detection) and "Detection of nucleic acids" (Bacterial pathogens identification) into separate sections for better readability. Include more information on the principles and techniques used for detecting each specific molecule, along with relevant references to support the claims.

5.      Conclusion: Summarize the main findings and contributions of the paper in a concise manner. Consider adding a future outlook section to discuss potential advancements and emerging trends in nanopore sensing techniques.

n.a

Author Response

  1. The paper "Detection of Biological Molecules Using Nanopore Sensing Techniques" provides an overview of the applications, methodological approaches, and challenges associated with nanopore sensing. Overall, the paper provides a good foundation on the topic of nanopore sensing. By addressing below suggestions, the paper can be further improved to enhance clarity and provide a more comprehensive understanding of the subject matter.

    1. Introduction:  Consider providing a clearer motivation for the importance of nanopore sensing in the introduction. Include more recent references (beyond [3]) to showcase the latest advancements in the field.

    We have revised the introduction to provide a more explicit and convincing rationale for the importance of nanopore detection. We now highlight how nanopore technology has emerged as a powerful tool to address a wide range of single-molecule level problems in pathologies for which conventional approaches are deficient. We believe that these revisions significantly enhance the introduction by providing a clearer motivation and showcasing the latest advancements in nanopore sensing.

    1. Applications in sensing and diagnostic: Provide more specific examples and references to support the claim that nanopores are the basis for the development of new technologies for rapid detection of various specific molecules involved in pathologies. Expand on the advantages and limitations of nanopore analysis methods compared to other techniques currently used for molecule detection.

    We have provided specific examples and references to highlight the applications of nanopores in sensing and diagnostics. We discuss how nanopores have been used to rapidly detect specific molecules. For instance, we mention studies where nanopores have been employed for the detection of protein and genetic markers. These examples are supported by recent references, including [15], [16], [17], [18].

    1. Methodological approaches and challenges:  Clearly define and explain key terms such as "translocation time" and "dwell time" to ensure better understanding for readers. Include specific details about the experimental parameters that can be adjusted to achieve high resolution in nanopore analysis methods. Provide additional references to support the discussion on bioinformatics analysis of the signal and the algorithms used for data analysis.

    We have provided clear definitions and explanations of key terms such as "translocation time" and "dwell time" to facilitate better comprehension for readers. We describe "translocation time" as the duration taken by a molecule to pass through the nanopore, while "dwell time" refers to the time a molecule spends within the nanopore during the analysis. To support the discussion on bioinformatics analysis of the signal and the algorithms used for data analysis, we have included additional references highlighting studies that have developed bioinformatics approaches and algorithms for the analysis of nanopore signals.

    1. Specific molecule detection: Organize the subsections under "Metabolic markers" (Glucose detection, Vitamin B1, Uric acid detection) and "Detection of nucleic acids" (Bacterial pathogens identification) into separate sections for better readability. Include more information on the principles and techniques used for detecting each specific molecule, along with relevant references to support the claims.

    We thank the reviewer for the suggestion, however, each molecule or category of metabolites are already grouped into sections and subsections (from 3.1 to 3.3 and, within each, 3.1.1 to 3.1.3 etc), Therefore, we kindly ask the reviewer to further detail on the suggestion so as to make the appropriate changes.

    1. Conclusion: Summarize the main findings and contributions of the paper in a concise manner. Consider adding a future outlook section to discuss potential advancements and emerging trends in nanopore sensing techniques.

    We have expanded the conclusion by including a Perspectives section in which we discuss emerging areas and the main improvements the nanopore sensing technique needs to reach high performance and potential directions for nanopore sensing research. We have also  highlighted the unique capabilities of nanopore sensing, such as single-molecule resolution, and real-time monitoring.

    Thank you once again for your constructive feedback, which has greatly improved our work quality.

Reviewer 2 Report

This review aims to highlight the versatility of the nanopore analysis method and, at the same time, provide insight into the challenges the method raises, based on the molecule to be studied. The paper includes a number of specific experimental aspects for various molecules with an important relevance for human health.

This manuscript is interesting, well fits with the scope of journal and is rather well written and interestingly addressed. Manuscript contributes to the field of bioanalytical chemistry and bioanalytical systems. Therefore, the manuscript eventually can be published after some minor improvements and corrections:

CRISPR-Cas-based bioanalytical systems recently are rather effective single-molecule detection bioanalytical systems, for this reason some reviews on the application of CRISPR/Cas system (The application of DNA polymerases and Cas9 as representative of DNA-modifying enzymes group in DNA sensor design (Review). Biosensors and Bioelectronics 2021, 175, 112867. // Towards application of CRISPR-Cas12a in the design of modern viral DNA detection tools (Review). Journal of Nanobiotechnology 2022, 20, 41.) could be taken into attention and overviewed in chapters ‘3.2.2. Genetic markers’ and ‘3.3. Protein detection’ and/or in some other parts of the manuscript where CRISPR-Cas-based systems are mentioned.

Importance of enzyme-based glucose sensors and increase of sensitivity by advancement of charge transfer efficiency (Charge transfer and biocompatibility aspects in conducting polymers based enzymatic biosensors and biofuel cells. Nanomaterials 2021, 11, 371.) could be mentioned in chapter ‘3.1.1. Glucose detection’.

Some more critical remarks on overviewed sensing systems could be provided in each subchapter advantages and disadvantages of overviewed sensing systems could be more efficiently addressed.

This review aims to highlight the versatility of the nanopore analysis method and, at the same time, provide insight into the challenges the method raises, based on the molecule to be studied. The paper includes a number of specific experimental aspects for various molecules with an important relevance for human health.

This manuscript is interesting, well fits with the scope of journal and is rather well written and interestingly addressed. Manuscript contributes to the field of bioanalytical chemistry and bioanalytical systems. Therefore, the manuscript eventually can be published after some minor improvements and corrections that re presented in 'Comments and Suggestions for Authors'.

Author Response

    1. CRISPR-Cas-based bioanalytical systems recently are rather effective single-molecule detection bioanalytical systems, for this reason some reviews on the application of CRISPR/Cas system (The application of DNA polymerases and Cas9 as representative of DNA-modifying enzymes group in DNA sensor design (Review). Biosensors and Bioelectronics 2021, 175, 112867. // Towards application of CRISPR-Cas12a in the design of modern viral DNA detection tools (Review). Journal of Nanobiotechnology 2022, 20, 41.) could be taken into attention and overviewed in chapters ‘3.2.2. Genetic markers’ and ‘3.3. Protein detection’ and/or in some other parts of the manuscript where CRISPR-Cas-based systems are mentioned.

    We appreciate your suggestion to include additional papers on the application of CRISPR-Cas systems in bioanalytical systems. As such, we have included those references in the section where CRISPR-Cas-based systems are discussed. Specifically, in '3.2.2. Genetic markers' we provide an overview of the applications of CRISPR-Cas systems in this context and reference the relevant papers you suggested.

    1. Importance of enzyme-based glucose sensors and increase of sensitivity by advancement of charge transfer efficiency (Charge transfer and biocompatibility aspects in conducting polymers based enzymatic biosensors and biofuel cells. Nanomaterials 2021, 11, 371.) could be mentioned in chapter ‘3.1.1. Glucose detection’.

    In Chapter 3.1.1, where we discuss glucose detection, we have incorporated the information suggested and emphasized the importance of charge transfer. Specifically, we reference the suggested article to support this discussion. We appreciate your valuable input and believe that this addition strengthens the chapter on glucose detection.

    1. Some more critical remarks on overviewed sensing systems could be provided in each subchapter advantages and disadvantages of overviewed sensing systems could be more efficiently addressed.

    As requested, we have included more critical remarks on the sensing systems presented within each subchapter and highlighted not only the advantages but also the limitations and challenges associated with each sensing system.

    Thank you once again for your constructive feedback, which has greatly improved our work quality.